# When Adversarial Training Meets Vision Transformers: Recipes from Training to Architecture

**Yichuan Mo**[1]  **Dongxian Wu**[2]  **Yifei Wang**[3]  **Yiwen Guo**[4]  **Yisen Wang**[1,5*]

[1] Key Lab. of Machine Perception (MoE),
School of Intelligence Science and Technology, Peking University
[2] The University of Tokyo
[3] School of Mathematical Sciences, Peking University
[4] Independent Researcher
[5] Institute for Artificial Intelligence, Peking University

## Abstract

Vision Transformers (ViTs) have recently achieved competitive performance in broad vision tasks. Unfortunately, on popular threat models, naturally trained ViTs are shown to provide no more adversarial robustness than convolutional neural networks (CNNs). Adversarial training is still required for ViTs to defend against such adversarial attacks. In this paper, we provide the *first and comprehensive study on the adversarial training recipe of ViTs* via extensive evaluation of various training techniques across benchmark datasets. We find that pre-training and SGD optimizer are necessary for ViTs' adversarial training. Further considering ViT as a new type of model architecture, we investigate its adversarial robustness from *the perspective of its unique architectural components*. We find, when randomly masking gradients from some attention blocks or masking perturbations on some patches during adversarial training, the adversarial robustness of ViTs can be remarkably improved, which may potentially open up a line of work to explore the architectural information inside the newly designed models like ViTs. Our code is available at `https://github.com/mo666666/When-Adversarial-Training-Meets-Vision-Transformers`.

## 1 Introduction

Recent years have witnessed the overwhelming advances of Vision Transformers (ViTs) [1, 2]. Different from widely deployed Convolutional Neural Networks (CNNs) [3, 4] that adopt a series of local convolutional operations on input images, ViTs adopt self-attention mechanisms [5] on a sequence of image patches. Based on large-scale pre-training, ViTs have achieved competitive and even better performance compared to CNNs in several fields such as semantic segmentation [6], object detection [7], and image generation [8]. Despite their great success on a growing number of vision tasks, ViTs fail in providing more adversarial robustness than CNNs on popular threat models [9, 10]. To defend against such adversarial examples crafted by adding human-imperceptible perturbations to images [11, 12], ViTs still demand adversarial training (adversarial training) [13–16], an approach that incorporates adversarial examples into training and obtains notable empirical robustness.

Notably, the implementation details of adversarial training make a big difference on CNNs backbones [17], which motivates us to comprehensively evaluate various training techniques for adversarial training of ViTs and thus provide the *first implementation benchmark for the training recipes of ViTs under adversarial training*. Since adversarial training is quite different from natural training, different insights are discovered, for example, pre-training is useful for adversarial robustness and

---

*Corresponding author: Yisen Wang (yisen.wang@pku.edu.cn).

SGD behaves better than AdamW for ViTs' adversarial training. Based on our empirical findings, we provide a practical recipe for adversarial training of ViTs to help researchers in their future work.

Although ViTs can achieve reasonable adversarial robustness via the above training recipes of adversarial training, it is unexplored whether or not we are able to further improve its robustness through the specific architectural information from ViTs, after all ViTs are a kind of model totally different from CNNs. Interestingly, when we randomly mask gradients from some attention blocks or mask perturbations on some patches during adversarial training, the robustness of ViTs can be further improved. We term these two simple methods as *Attention Random Dropping* (ARD) and *Perturbation Random Masking* (PRM) respectively, and conduct extensive experiments to verify their effectiveness.

Our main contributions are summarized as follows:

- We comprehensively evaluate training techniques for ViTs under the setting of adversarial training on popular threat models, and discover many interesting insights that are different from the natural training scenario.
- Based on the unique architectural information from ViTs, we propose *Attention Random Dropping* (ARD) and *Perturbation Random Masking* (PRM) as a warming-up strategy to improve the adversarial robustness of ViTs. In particular, ARD randomly masks gradients from some attention blocks while PRM randomly masks perturbations on some patches.
- Our work not only provides the first implementation benchmark (bags of tricks) for adversarially trained ViTs, but also reminds researchers of the potential of architectural information inside the new brand of models like ViTs.

## 2 Related Work

### 2.1 Training Strategies for ViTs

Due to the lack of inductive bias [1], ViTs require more training techniques to reach their performance potential. Dosovitskiy et al [1] first pretrained ViTs on the large-scale datasets (ImageNet-21k or JFT-300M [18]) and achieved competitive performance on downstream tasks. Touvron et al [19] applied strong data augmentations such as Randaugment [20] and Mixup [21] to train ViTs from scratch. They also find AdamW [22] performs better than SGD [23] for training ViTs. Other studies [1, 24, 25] adopted gradient clipping to stabilize and accelerate the convergence of ViTs. Steiner et al [26] conducted a comprehensive study on data augmentation and model regularization of ViTs to improve their natural accuracy. For the robustness of ViTs, the training strategies, especially under adversarial training, are not fully investigated yet. That is exactly what we are doing in this paper.

### 2.2 Adversarial Robustness of ViTs

Given a clean example $x$ with its class label $y$ and a model $f_\theta(\cdot)$ with its parameters $\theta$, the goal of an adversary is to find an adversarial perturbation $\delta$ that fools the network into making an incorrect prediction for the perturbed image (*i.e.*, $f_\theta(x + \delta) \neq y$), while the perturbation norm does not exceed $\epsilon$ (*i.e.*, $\|\delta\|_\infty \leq \epsilon$),

$$\delta = \arg\max_{\|\delta\|_\infty \leq \epsilon} \mathcal{L}(f_\theta(x + \delta), y), \tag{1}$$

where $\mathcal{L}(\cdot, \cdot)$ is usually the cross entropy loss. Previous studies [12, 27, 13] proposed several methods to generate adversarial examples, and Project Gradient Descent (PGD) [13] becomes the most popular one:

$$\delta \leftarrow \Pi_\epsilon\big(\delta + \alpha \cdot \text{sign}\big(\nabla_\delta \mathcal{L}(f_\theta(x + \delta), y)\big)\big). \tag{2}$$

To defend against adversarial attacks, adversarial training (adversarial training) and its variants [13, 28, 14–16, 29] become the most promising and effective method [30–32], which directly incorporates adversarial examples into training:

$$\theta = \arg\min_\theta \frac{1}{N} \sum_{i=1}^{N} \max_{\|\delta_i\|_\infty \leq \epsilon} \mathcal{L}(f_\theta(x_i + \delta_i), y_i). \tag{3}$$

Despite extensive understandings of adversarial robustness on CNNs, only a few studies focus on the robustness of ViTs, which can be classified into the following categories: 1) naturally trained

(non-adversarially trained) ViTs: In [33, 9, 34, 10], they revealed that naturally trained ViTs are more robust than CNNs against extremely small perturbations (*e.g.*, under threat model of $\epsilon = 0.001$ on ImageNet, which is much smaller than the popular threat model of $\epsilon = 4/255 \approx 0.0157$); 2) adversarially trained ViTs: In [35, 9], they demonstrated that adversarially trained ViTs fail in providing more adversarial robustness on popular threat models compared to CNNs.

However, these previous works all lack a comprehensive study on the adversarial training process of ViTs, which may result in an incomplete and biased impression on the robustness of adversarially trained ViTs. Therefore, under the setting of adversarial training of ViTs, we for the first time comprehensively evaluate bags of training tricks across benchmark datasets, and provide a whole adversarial training recipe for ViTs to obtain remarkable robustness.

## 3 Strategies for Adversarial Training of ViTs

At the first glance, we may think there are already many works studying the training strategies of ViTs. The BIG difference from them lies on the training paradigm: previous research focuses more on the natural training of ViTs while here we focus on ViTs' adversarial training. Since adversarial training is quite different from natural training, different insights might be discovered.

In this section, we comprehensively evaluate several training techniques in terms of data (pre-training and data augmentation) and training (optimizer, learning rate schedule, and gradient clipping) across various architectures (vanilla ViT and Swin) and datasets (CIFAR-10 and Imagenette), so as to provide the first training recipe for adversarial training of ViTs.

**Datasets.** Since adversarial training is time-consuming, small datasets remain popular for adversarial robustness [32]. Here, we use datasets of CIFAR-10 [36] and Imagenette [37] (a subset of 10 classes from ImageNet-1K). Note that the latest version of Imagenette (imagenette-v2[2]) reshuffles the sampled subset of ImageNet-1K and then splits the training and validation set. This leads to an overlapping issue between the pre-training set (ImageNet-1K) and the reshuffled validation set. To avoid the leakage of validation data during pre-training, we select the previous version of Imagenette (imagenette-v1[3]).

**Architectures.** We use the vanilla ViT (ViT-B) [1] and Swin-B [2] as baseline models. ViT first successfully introduced Transformers from natural language processing to computer vision and achieved competitive recognition performance compared to CNNs. With the help of multi-stage hierarchical architecture, Swin became a general-purpose backbone and achieved SOTA performance on several downstream tasks. By default, all models are pretrained on ImageNet-1K[4]. To accommodate the smaller image size of CIFAR-10, we downsample the patch embedding kernel from $16 \times 16$ to $4 \times 4$.

**Threat Models.** We apply the commonly-used $\ell_\infty$ attack ($\|\delta\|_\infty \leq \epsilon$) and set $\epsilon = 8/255$ for CIFAR-10 and Imagenette. When evaluating the adversarial robustness, we apply 20-step PGD (PGD-20) [13] with the step size $2/255$ and Auto-Attack (AA) [31] that is the strongest attack to verify the empirical robustness via an ensemble of diverse attacks (*i.e.*, two variants of PGD attack, FAB attack [38], and Square Attack [39]).

**Training Settings.** All models (unless otherwise specified) are pre-trained on ImageNet-1K and are adversarially trained for 40 epochs using SGD with weight decay 1e-4, and an initial learning rate 0.1 that is divided by 10 at the 36-th and 38-th epoch. Simple data augmentations such as random crop with padding and random horizontal flip are applied. During adversarial training, we use PGD-10 with step size $2/255$ to craft adversarial examples.

### 3.1 The Usefulness of (Natural) Pre-training

The success of ViTs heavily relies on natural pre-training on large datasets [1]. Meanwhile, natural pre-training is commonly believed to be useless for improving the adversarial robustness of naturally trained ViTs [9]. A question is naturally raised here: under *adversarial training of ViTs*, whether natural pre-training is useful or not.

---

[2]https://s3.amazonaws.com/fast-ai-imageclas/imagenette2.tgz
[3]https://s3.amazonaws.com/fast-ai-imageclas/imagenette.tgz
[4]https://github.com/rwightman/pytorch-image-models

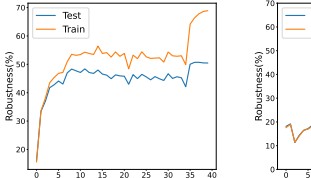 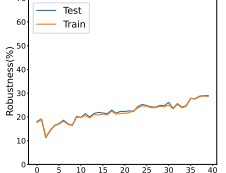 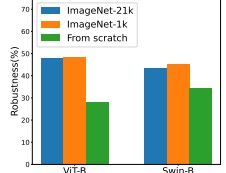 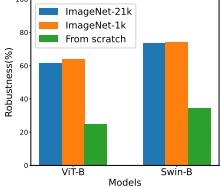

(a) Adversarial training curve of ViT-B on CIFAR-10 (*Left*: with pre-training, *Right*: training from scratch)

(b) Final robustness of different pre-training strategies (*Left*: CIFAR-10, *Right*: Imagenette)

Figure 1: Effects of different pre-training strategies on adversarial training of ViTs.

We first plot the learning curve of ViT-B with and without pre-training. Specifically, one model is updated from an initialization from natural pre-training on ImageNet-1K, while the other is from random initialization. In Figure 1(a), we find that ViT without pre-training achieves low robustness because under-fitting on the training data hinders the performance (the training robustness is only $\sim$ 30%). Meanwhile, ViT with pre-training fits training data better and improves robustness by a notable margin (+20%). This suggests that the current random initialization for weight parameters of ViTs is not suitable for adversarial training, even on small datasets like CIFAR-10, while natural pre-training on large datasets is useful for successful adversarial training of ViTs. This conclusion seems different from the previous research [9]. This is because, although both study adversarial robustness, we focus on the effects of natural pre-training for adversarially trained ViTs, while they focus more on the robustness of naturally trained models.

Further, we explore if a larger dataset can bring more robustness. Before adversarial training on CIFAR-10, we adopt 3 pre-training strategies respectively: 1) natural pre-training on ImageNet-21K (an extremely large dataset with 14M images from 21K classes), 2) natural pre-training on ImageNet-1K (a large dataset with more than 1.2M images from 1K classes), and 3) no pre-training (*i.e.*, adversarial training from scratch). The final robustness under AA is shown in Figure 1(b). Compared to adversarial training from scratch, pre-training on both ImageNet-1K and ImageNet-21K can improve the robustness of ViTs by a notable margin ($\sim 16\%$ on average), while ImageNet-21K fails in providing higher robustness (it even slightly hurts the performance) compared to ImageNet-1K. Similar trends can also be observed on Imagenette.

In conclusion, natural pre-training is necessary for ViTs to fit the training data during adversarial training, while a larger dataset is unable to provide better robustness.

## 3.2 The Necessity of Gradient Clipping

Table 1: The performance (%) of adversarially trained ViT, Swin, and DeiT with or without (w/o) gradient clipping.

| Model | CIFAR-10 | | Imagenette | |
|---|---|---|---|---|
| | Natural | AA | Natural | AA |
| ViT-B | **85.53** | **48.33** | **91.40** | **64.20** |
| ViT-B w/o GC | 10.00 | 10.00 | 10.00 | 10.00 |
| Swin-B | **83.75** | **45.28** | **95.60** | **74.00** |
| Swin-B w/o GC | 82.42 | 44.90 | 95.00 | 73.00 |
| DeiT-S | **84.13** | **47.26** | **90.40** | **60.20** |
| DeiT-S w/o GC | 10.00 | 10.00 | 10.00 | 10.00 |

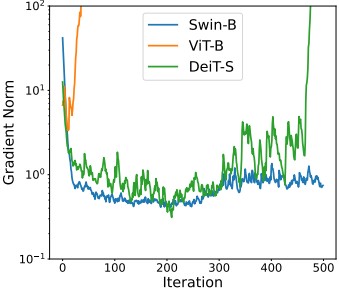

Figure 2: The $\ell_2$-norm of gradients during adversarial training of ViTs.

In natural training, some Transformers implementations (*e.g.*, ViTs and Swin) apply gradient clipping, while the others (*e.g.*, DeiT) do not. Here, we explore whether it is necessary to use gradient clipping during adversarial training. First, we find that, without gradient clipping, it is difficult to adversarially train ViT or DeiT on small datasets. For example, the final robustness of ViT-B and DeiT-S is only 10% on both CIFAR-10 and Imagenette in Table 1. To study the difficulty, we visualize the norm of gradient during adversarial training in Figure 2, and find the gradient explodes at the very beginning of training. Then we adopt gradient clipping [40] again to adversarial training of ViTs. Specifically, we

ensure the maximum of gradient vector is 1 under the $l_2$-norm. In Table 1, with the help of gradient clipping, ViT-B (DeiT-S) achieves the robustness of 48.33% (47.26%) and 64.20% (60.20%) on CIFAR-10 and Imagenette. Although Swin-B can be adversarially trained directly, gradient clipping still improves the performance by a notable margin. In comparison, gradient clipping has almost no effect on the robustness of CNNs, as shown in Appendix B. In summary, for ViTs, although gradient clipping is optional in natural training, it seems necessary for their adversarial training.

## 3.3 The Effect of More Training Epochs

The natural training of ViTs usually requires more epochs for better natural accuracy (*e.g.*, 300 epochs for Swin [2] on ImageNet-1K) compared to CNNs (*e.g.*, 90 epochs for ResNet [4] on ImageNet-1K). Adversarial training of ViTs is then supposed to require longer training epochs for better adversarial robustness [9]. Here, we further investigate it by adversarially training ViTs for 80 epochs (the default setting is 40 epochs) with the learning rate decay at the 75-th and 78-th epoch. Figure 3(a) illustrates that neither ViT-B nor Swin-B benefits from the longer training, which potentially hurts the final robustness instead.

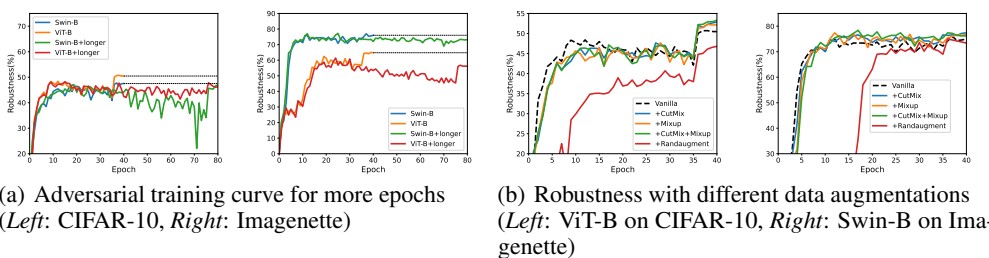

(a) Adversarial training curve for more epochs (*Left*: CIFAR-10, *Right*: Imagenette)

(b) Robustness with different data augmentations (*Left*: ViT-B on CIFAR-10, *Right*: Swin-B on Imagenette)

Figure 3: Learning curves of adversarial training with longer training epochs or advanced data augmentations. The robustness is evaluated under PGD-20.

## 3.4 The Effect of Advanced Data Augmentations

In adversarial training of CNNs, a majority of advanced data augmentations like Mixup [21] and Randaugment [20] are proven to be unsuccessful (without considering model weight averaging) [41, 42], which is contrary to natural training. Here, we explore if these advanced data augmentations can improve robustness for adversarial training of ViTs. Specifically, we incorporate CutMix [43], Mixup [21], and Randaugment [20] to adversarial training of ViTs respectively. In Figure 3(b), on CIFAR-10, ViT-B obtains higher robustness with CutMix or Mixup or both, while behaving worse with Randaugment. We also find similar phenomena of Swin-B on Imagenette. This is because Randaugment is too difficult for adversarial training of ViTs. In conclusion, a suitable combination of data augmentations can directly improve the adversarial robustness of ViTs, which is different from CNNs (needing model weight averaging). In the following experiments, we always adopt CutMix and Mixup to adversarial training of ViTs by default.

Table 2: The performance (%) of adversarially trained ViT-B and Swin-B by different optimizers and learning rate schedulers.

| Dataset | Optimizer | ViT-B | | Swin-B | |
|---|---|---|---|---|---|
| | | Natural | AA | Natural | AA |
| CIFAR-10 | AdamW+cyclic | 78.67 | 46.16 | 78.33 | 45.20 |
| | AdamW+piecewise | 80.76 | 46.76 | 10.00 | 10.00 |
| | SGD+cyclic | 83.06 | 48.91 | 81.83 | 45.46 |
| | SGD+piecewise | **83.16** | **49.06** | **83.36** | **46.89** |
| Imagenette | AdamW+cyclic | 92.00 | 66.80 | 93.40 | 72.40 |
| | AdamW+piecewise | 87.40 | 59.40 | 10.00 | 10.00 |
| | SGD+cyclic | 93.20 | 66.60 | 95.40 | 74.40 |
| | SGD+piecewise | **93.40** | **67.00** | **96.40** | **74.60** |

### 3.5 The Effect of Optimizer and Learning Rate Scheduler

Recalling that SGD achieves the best adversarial robustness in CNNs [13, 41]. By contrast, the default optimizer of ViTs (natural training) is AdamW [22]. Here, we investigate what is the appropriate optimizer for adversarial training of ViTs in order to obtain better robustness. We adversarially train models based on every combination of the optimizer[5] (AdamW and SGD) and the learning rate schedule (cyclic and piecewise). The results are summarized in Table 2. If fixing the learning rate scheduler, whatever cyclic or piecewise, SGD almost all outperforms AdamW in terms of adversarial robustness. When fixing the optimizer, we find that cyclic learning rate scheduler is suitable for AdamW while the piecewise learning rate scheduler is a better choice for SGD. Furthermore, SGD+piecewise achieves better robustness compared to AdamW+cyclic in all datasets and models, *i.e.*, on CIFAR-10, SGD + piecewise helps ViT-B improve the natural accuracy by 4.49% and the adversarial robustness by 2.90% over AdamW + cyclic. In summary, the combination of SGD and piecewise learning rate scheduler may be a good choice for adversarial training of ViTs.

### 3.6 Insights on the Adversarial Training of ViTs

With the above comprehensive evaluation of different training techniques, we can provide a practical adversarial training recipe for ViTs. First, pre-training and gradient clipping are almost necessary. Second, it would be better if incorporating advanced data augmentations (*e.g.*, Cutmix and Mixup) and adopting SGD optimizer with piecewise learning rate scheduler. Third, there seems no need to train longer epochs for the robustness improvement.

Table 3: The comparison of adversarially trained ViTs and CNNs.

(a) CIFAR-10 and Imagenette

| Model | ResNet50 | Swin-Ti | WRN-50-2 | Swin-S |
|---|---|---|---|---|
| #Parameters | 22.5M | 27.5M | 66.9M | 48.8M |
| CIFAR-10 (AA) | **49.02** | 43.98 | **51.33** | 44.88 |
| Imagenette (AA) | 61.00 | **71.80** | 62.00 | **73.80** |

(b) ImageNet-1K

| Model | WRN-50-2 | Swin-B |
|---|---|---|
| Natural | 68.46 | **74.36** |
| AA | 38.14 | **38.61** |

To show the potential of ViTs' adversarial training, we adversarially train ViTs following the above recipe and CNNs following the default setting, ensuring both of them own similar numbers of parameters. In Table 3(a), on CIFAR-10, similar to previous findings [9], ViTs indeed do not advance the adversarial robustness compared to CNNs, while we think that we should not be overly pessimistic about this, after all CIFAR-10 is a low-resolution dataset, while ViTs are usually trained on high-resolution datasets. We might need a better adaption method for this input size difference. On the positive side, on high-resolution datasets like ImageNette (Table 3(a)) and ImageNet-1K (Table 3(b)), the robustness results are totally different. On Imagenette, transformers of different sizes (Swin-Ti for small size, and Swin-S for medium size) outperform their competitors (ResNet50 or WideResNet-50-2 respectively) with a similar even smaller number of parameters by a notable margin ($> 10\%$). On ImageNet-1K (a more difficult dataset for adversarial tasks), we achieve robustness of 38.61% [6], which is higher than the current state-of-the-art result of 38.14% on RobustBench . This indicates the great potential for ViTs in terms of adversarial training.

## 4 An Architectural Perspective for Adversarially Robust ViTs

In contrast to CNNs, ViTs are a completely different architecture. For example, it divides images into sequences of patches before feeding into the model, and it directly integrates global information across the entire images even in the lowest layers using self-attention [1]. This fundamental difference naturally raises a question: whether or not we are able to further improve its robustness based on such specific architectural information. In this section, we explore the impact of these two most obvious architectural changes (multi-head attention and patch-based image splitting) inside ViTs on the adversarial robustness.

---

[5]AdamW with initial learning rate 5e-4 and weight decay 0.3 and SGD with the default setting in Section 3
[6]For the experimental details, please refer to Appendix D.

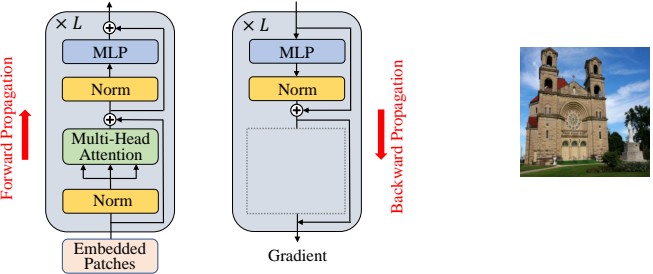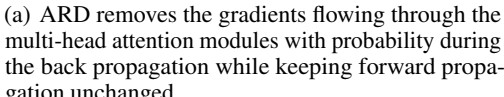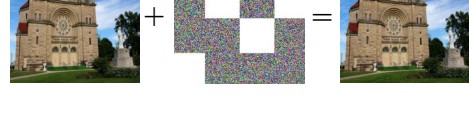

(a) ARD removes the gradients flowing through the multi-head attention modules with probability during the back propagation while keeping forward propagation unchanged.

(b) PRM randomly masks a proportion of perturbation with probability during the forward propagation while keeping backward propagation unchanged.

Figure 4: The diagram of our proposed method to improve the adversarial robustness of ViTs.

## 4.1 Attention Random Droppping (ARD)

In ViTs, embedded patches are repeatedly processed by encoder blocks, any of which consist of a multi-head attention and a MLP layer, as shown in Figure 4(a). For the $i$-th block, its input $z_{i-1}$ is first processed by a multi-head attention $\mathcal{A}^{(i)}(\cdot)$, and intermediate result $z'_i = \mathcal{A}^{(i)}(z_{i-1}) + z_{i-1}$ is further processed by the remaining parts $f^{(i)}(\cdot)$ (including a MLP layer and a skip connection). For a ViT with $L$ blocks, its output of the last (*i.e.*, $L$-th) block is

$$z_L = f^{(L)}(z'_L) = f^{(L)}(\mathcal{A}^{(L)}(z_{L-1}) + z_{L-1}). \tag{4}$$

According to the chain rule in calculus, the gradient of a loss function $\mathcal{L}$ with respect to the input of the last block $z_{L-1}$ can then be decomposed as

$$\frac{\partial \mathcal{L}}{\partial z_{L-1}} = \frac{\partial \mathcal{L}}{\partial z_L} \frac{\partial z_L}{\partial z_{L-1}} = \frac{\partial \mathcal{L}}{\partial z_L} \frac{\partial f^{(L)}}{\partial z'_L} \left( \frac{\partial \mathcal{A}^{(L)}}{\partial z_{L-1}} + 1 \right). \tag{5}$$

Here, we propose the *Attention Random Dropping* (ARD), which randomly drops the gradient flowing the multi-head attention $\mathcal{A}^{(L)}(\cdot)$,

$$\frac{\partial \mathcal{L}}{\partial z_{L-1}} = \frac{\partial \mathcal{L}}{\partial z_L} \frac{\partial f^{(L)}}{\partial z'_L} \left( u_L \cdot \frac{\partial \mathcal{A}^{(L)}}{\partial z_{L-1}} + 1 \right), \tag{6}$$

where we set $u_L$ to 0 with the probability of $p$ and to 1 otherwise. We continue to use the chain rule to obtain the gradient of ARD with respect to the input image $x$,

$$\frac{\partial \mathcal{L}}{\partial x} = \frac{\partial \mathcal{L}}{\partial z_L} \left( \prod_{i=1}^{L} \frac{\partial f^{(i)}}{\partial z'_i} \left( u_i \cdot \frac{\partial \mathcal{A}^{(i)}}{\partial z_{i-1}} + 1 \right) \right) \frac{\partial z_0}{\partial x}, \tag{7}$$

where $z_0$ is the input embedded patches and all random variables $u_i$ are *i.i.d.* Figure 4(a) also illustrates the backward propagation using ARD.

When ARD meets adversarial training, we apply ARD as a warming-up strategy. After the warming-up periods, the training becomes vanilla adversarial training. In particular, the parameter $p$ in ARD decreases linearly from 1 to 0 during the first $n_w$ epochs, *i.e.*, $p = 1 - n/n_w$ where $n$ is the current epoch ($n \le n_w$). Here, $p$ is a dynamic parameter along with the training process. At the beginning, relatively weak adversarial samples (large $p$) are used to warm up while as the training continues, the probability $p$ should be gradually smaller to 0 to ensure the strength of the generated adversarial samples during the whole adversarial training process.

## 4.2 Perturbation Random Masking (PRM)

ViTs always split input images into non-overlapping patches, and then process them further. Considering this patch-based image splitting, we introduce the *Perturbation Random Masking* (PRM).

Specifically, the perturbation on a patch will be masked with probability $k$ during adversarial example generation:

$$\delta' \leftarrow M \cdot \delta,$$
$$\delta \leftarrow \Pi_\epsilon \left( \delta' + \alpha \cdot \text{sign} \left( \frac{\partial \mathcal{L}(f(x + \delta'), y)}{\partial \delta'} \right) \right). \tag{8}$$

where $\delta$ is the adversarial perturbation, and $M$ is the mask to remove the perturbations on some patches. The second equation is exactly the update rule of PGD attack.

Similar to ARD, we incorporate PRM as a warming-up strategy. Considering an image that is split into $J$ patches, the perturbations on $\lfloor J \cdot k \rfloor$ patches will be masked for each iteration during adversarial example generation within adversarial training, and we linearly decrease $k$ from 1 to 0 at the first $n_w$ epochs, i.e., $k = 1 - n/n_w$ where $n$ is the current epoch ($n \leq n_w$). Here, same as $p$, $k$ is also a dynamic parameter along with the training process.

### 4.3 Overall Algorithm

Obviously, ARD and PRM are two individual methods that have impact on two different aspects, i.e., ARD influences the backward propagation while PRM influences the update of adversarial perturbations. Thus, we can easily combine them as a mixed warming-up strategy:

$$\delta' \leftarrow M \cdot \delta,$$
$$\delta \leftarrow \Pi_\epsilon \left( \delta' + \alpha \cdot \text{sign} \left( \frac{\partial \mathcal{L}}{\partial z_L} \left( \prod_{i=1}^{L} \frac{\partial f^{(i)}}{\partial z_i'} (u_i \frac{\partial \mathcal{A}^{(i)}}{\partial z_{i-1}} + 1) \right) \frac{\partial z_0}{\partial \delta'} \right) \right). \tag{9}$$

The details of ARD and PRM based adversarial training for ViTs are summarized in Appendix C.

### 4.4 Evaluation on Benchmark Datasets

In this section, we evaluate the effectiveness of ARD and PRM for the adversarial robustness of ViTs on benchmark datasets.

**Experimental Settings.** For models, we apply 4 kinds of ViTs (vanilla ViT [1], DeiT [19], ConViT [44], and Swin [2]) and 3 kinds of scales (base, small, and tiny) for each respectively. For robustness evaluation, we also adopt 20 steps CW [27] ($\ell_\infty$ version of CW loss optimized by PGD-20) and 100 steps PGD$_\infty$[13] in addition to PGD-20 and AA. We evaluate the performance on: 1) the vanilla setting (the basic setting determined in Section 3), 2) the setting with ARD, 3) the setting with PRM, and 4) the setting with ARD and PRM. To reveal the potential of our proposed methods, we select the hyperparameter $n_w$ via a grid search over $\{5, 10, 15, 20\}$ for each combination of methods (+ARD, +PRM, or +both) and architectures (ViT-S, ViT-B, DeiT-Ti, *et al.*).

**Experimental Results.** In Table 4, after comparing the same kind of ViTs with different scales, we observe that increasing the size of ViTs can improve both the accuracy on clean examples and the robustness against adversarial attacks. Interestingly, the better architecture in natural training on the high-resolution dataset (*e.g.*, Swin on ImageNet-1K) fails to lead to better robustness in adversarial training on the low-resolution dataset. We conjecture the inductive bias (multi-scale objects and the high resolution of pixels on high-resolution images) inside Swin is no longer valid on low-resolution images, which results in unsatisfactory performance. Moreover, diving into each ViT model with different architectural approaches, we find that our proposed ARD or PRM has already boosted the robustness and accuracy, and the combination of both methods can improve the performance further. For example, compared to the vanilla adversarial training which achieves the robustness of 44.88% with Swin-S on CIFAR-10, we increase the robustness to 46.04% (+ 1.16%) using ARD or 46.01% (+ 1.13%) using PRM, while we further improve the performance to 46.17% (+ 1.29%) using their combination. Similar results are also observed on large datasets (*i.e.* ImageNet-1k) in Appendix D.

### 4.5 Combination with Other Defense Methods

Although the above discussions are mainly concentrated on standard adversarial training, we further demonstrate that our methods: ARD and PRM, can be easily combined with other stronger defense methods. We adopt TRADES [14] and MART [15] here because of their outstanding performance evaluated in [32]. The settings are the same as in Section 4.4 except for two points: 1) we do not apply

Table 4: Performance (%) of ARD and PRM with different ViT variants on benchmark datasets. Note that 'B' denotes base, 'S' denotes 'small', and 'Ti' denotes 'tiny'. Here we do not report the results of ViT-Ti and DeiT-B because ViT-Ti does not exist and the architecture of DeiT-B is the same as ViT-B. The best results are in **bold**.

| Model | Method | CIFAR-10 | | | | | Imagenette | | | | |
|---|---|---|---|---|---|---|---|---|---|---|---|
| | | Natural | CW-20 | PGD-20 | PGD-100 | AA | Natural | CW-20 | PGD-20 | PGD-100 | AA |
| ViT-S | vanilla | 79.59 | 48.22 | 50.86 | 50.73 | 46.37 | 90.40 | 64.00 | 63.80 | 63.00 | 62.80 |
| | +ARD | 81.70 | 49.07 | 51.72 | 51.42 | 47.12 | 91.40 | 64.20 | 64.60 | 64.20 | 62.80 |
| | +PRM | 81.77 | 49.03 | 51.67 | 51.46 | 47.22 | **92.00** | **64.60** | 64.60 | 64.20 | **63.80** |
| | +both | **81.86** | **49.09** | **51.73** | **51.46** | **47.33** | 91.40 | 64.20 | **65.20** | **64.60** | 63.00 |
| ViT-B | vanilla | 83.16 | 51.11 | 52.98 | 52.71 | 49.06 | 93.40 | 68.00 | 68.80 | 68.00 | 67.00 |
| | +ARD | 84.21 | 51.61 | 53.41 | 53.10 | 49.65 | 94.80 | 69.40 | 68.60 | 68.20 | 68.20 |
| | +PRM | 84.31 | 52.16 | 53.79 | 53.47 | 50.01 | 94.40 | 70.60 | 69.80 | 69.40 | 69.20 |
| | +both | **84.90** | **52.27** | **53.80** | **53.51** | **50.03** | **95.00** | 70.60 | **70.00** | **69.60** | **69.60** |
| DeiT-Ti | vanilla | 75.46 | 45.40 | 48.10 | 47.96 | 43.62 | 82.40 | 55.60 | 56.00 | 55.80 | 53.80 |
| | +ARD | 77.95 | 47.28 | 49.41 | 49.21 | 45.45 | 87.60 | 63.60 | 63.40 | 63.20 | 62.80 |
| | +PRM | 79.09 | 47.64 | 49.76 | 49.53 | 45.68 | 89.20 | 63.40 | 62.80 | 62.60 | 61.80 |
| | +both | **79.60** | **48.04** | **50.33** | **50.15** | **45.99** | **90.20** | **64.80** | **64.00** | **64.00** | **63.00** |
| DeiT-S | vanilla | 81.43 | 49.53 | 51.88 | 51.75 | 47.40 | 92.20 | 64.20 | 64.60 | 63.40 | 63.40 |
| | +ARD | 81.76 | 49.49 | 51.65 | 51.43 | 47.55 | 90.80 | 67.00 | 66.00 | 66.00 | 65.80 |
| | +PRM | 83.02 | 50.47 | 52.50 | 52.26 | 48.27 | 91.00 | 66.20 | 65.80 | 65.40 | 65.00 |
| | +both | **83.04** | **50.52** | **52.52** | **52.36** | **48.34** | **91.00** | **67.00** | **66.60** | **66.20** | **65.80** |
| ConViT-Ti | vanilla | 53.09 | 30.87 | 33.63 | 33.61 | 29.65 | 63.60 | 37.40 | 39.20 | 39.20 | 36.60 |
| | +ARD | 79.87 | **47.54** | **50.14** | **49.90** | **45.63** | 90.20 | 64.00 | 63.60 | 63.40 | 63.20 |
| | +PRM | 76.78 | 45.55 | 48.14 | 47.98 | 43.60 | 90.40 | 64.40 | 63.80 | 63.40 | 63.40 |
| | +both | **80.28** | 47.47 | 49.86 | 49.55 | 45.42 | **90.40** | **65.40** | **65.00** | **64.80** | **64.40** |
| ConViT-S | vanilla | 54.03 | 31.73 | 34.61 | 34.60 | 30.60 | 87.40 | 62.80 | 64.20 | 63.40 | 61.60 |
| | +ARD | 84.06 | 50.83 | 52.72 | 52.44 | 48.71 | 94.00 | 68.40 | 67.80 | 67.60 | 67.20 |
| | +PRM | 84.05 | 50.79 | 52.96 | 52.56 | 48.72 | 92.80 | 67.60 | 67.40 | 67.20 | 67.00 |
| | +both | **84.32** | **50.94** | **53.10** | **52.81** | **48.85** | **94.40** | **68.80** | **68.20** | **67.80** | **67.60** |
| ConViT-B | vanilla | 61.54 | 35.63 | 38.77 | 38.71 | 34.21 | 92.20 | 69.20 | 68.20 | 68.20 | 68.00 |
| | +ARD | 85.36 | **51.51** | 53.16 | 52.96 | 49.16 | 93.80 | 70.60 | 71.00 | 70.20 | 69.60 |
| | +PRM | 85.48 | 51.48 | 52.83 | 52.48 | 49.28 | 94.20 | 70.00 | 69.40 | 69.40 | 69.20 |
| | +both | **85.80** | 51.47 | **53.36** | **53.05** | **49.33** | **95.20** | **72.60** | **73.00** | **72.20** | **70.60** |
| Swin-Ti | vanilla | 79.34 | 45.74 | 47.95 | 47.75 | 43.98 | 94.80 | 72.80 | 72.80 | 72.40 | 71.80 |
| | +ARD | 78.52 | 45.70 | 48.00 | 47.91 | 43.84 | 95.60 | 74.40 | 74.20 | 73.60 | 73.40 |
| | +PRM | 81.94 | 46.93 | 48.56 | 48.35 | 44.91 | 96.20 | 74.40 | 74.40 | 73.80 | 73.40 |
| | +both | **82.63** | **47.61** | **48.87** | **48.62** | **45.31** | **96.20** | **74.60** | **74.40** | **74.20** | **74.20** |
| Swin-S | vanilla | 79.34 | 46.56 | 48.53 | 48.32 | 44.88 | 95.40 | 74.60 | 74.00 | 74.00 | 73.80 |
| | +ARD | 82.07 | 47.84 | 49.56 | 49.31 | 46.04 | 96.00 | 75.60 | 75.00 | 74.80 | 74.60 |
| | +PRM | 84.24 | 48.17 | 49.63 | 49.38 | 46.01 | 96.00 | 75.60 | 75.00 | 74.60 | 74.40 |
| | +both | **84.46** | **48.52** | **50.02** | **49.66** | **46.17** | **96.00** | **76.00** | **75.00** | **75.00** | **74.80** |
| Swin-B | vanilla | 83.36 | 48.22 | 50.19 | 49.88 | 46.89 | 96.40 | 76.80 | 75.80 | 75.20 | 74.60 |
| | +ARD | 81.24 | 47.64 | 49.19 | 48.83 | 44.38 | 97.00 | 78.00 | 77.20 | 76.40 | 75.80 |
| | +PRM | 84.07 | 49.68 | 50.95 | 50.66 | 47.25 | 96.80 | 77.40 | 76.20 | 76.00 | 75.80 |
| | +both | **84.16** | **49.78** | **51.47** | **51.19** | **47.50** | **97.20** | **78.00** | **77.40** | **77.20** | **76.20** |

CutMix and Mixup for data augmentations to keep in line with their original papers, and 2) we always set $n_w$ as 10 for simplicity. The experiments are conducted on DeiT-Ti, ConViT-Ti, and Swin-Ti. As shown in Table 5, the robustness and accuracy of TRADES and MART are improved on both datasets. For example, for DeiT-Ti on CIFAR-10, AA robustness increases by 0.77% on TRADES and 1.63% on MART. It demonstrates that ARD and PRM can further improve the performance of existing defense methods.

### 4.6 Hyperparameter Analysis

For simplicity of our proposed warming-up strategy, we use the same hyperparameter $n_w$ as the number of warming-up epochs for both ARD and PRM. Taking CIFAR-10 as an example, we firstly analyze this shared hyperparameter $n_w \in \{0, 5, 10, 15, 20, 25\}$ on four ViT variants (ViT, DeiT, ConViT, and Swin), and the settings are the same as Section 4.4. AA is used to evaluate the robustness. In Figure 5(a), the robustness is significantly improved as long as adopting our proposed warming-up strategy into adversarial training, *i.e.*, $n_w > 0$. We obtain the highest robustness at the best $n_w$ (*e.g.*, $n_w = 15$ for Swin-S). Besides, the improvements are not sensitive to the hyperparameter, which allows us to tune $n_w$ easily.

To further exploit the proposed method, we perform experiments on Swin-S using two individual hyperparameters $n_w^a$ and $n_w^p$ for ARD and PRM respectively. When fixing one (*e.g.*, $n_w^a$) as 15 (the

Table 5: Performance (%) of our proposed ARD and PRM when combined with other defense methods. The best results are in **bold**.

| Model | Method | CIFAR-10 | | | | | Imagenette | | | | |
|---|---|---|---|---|---|---|---|---|---|---|---|
| | | Natural | CW-20 | PGD-20 | PGD-100 | AA | Natural | CW-20 | PGD-20 | PGD-100 | AA |
| DeiT-Ti | TRADES | 78.70 | 46.78 | 49.63 | 49.58 | 46.25 | 88.00 | 63.00 | 62.60 | 62.40 | 61.20 |
| | +Ours | **80.24** | **47.60** | **51.02** | **50.97** | **47.02** | **89.00** | **63.20** | **64.40** | **64.00** | **61.80** |
| | MART | 71.7 | 45.95 | 49.52 | 49.37 | 44.34 | 80.40 | 55.40 | 56.20 | 56.00 | 52.60 |
| | +Ours | **74.89** | **47.60** | **51.18** | **51.16** | **45.97** | **86.40** | **61.80** | **63.40** | **63.20** | **62.20** |
| ConViT-Ti | TRADES | 77.70 | 45.09 | 48.71 | 48.63 | 44.65 | 83.80 | 58.80 | 60.40 | 60.20 | 57.80 |
| | +Ours | **80.02** | **47.33** | **50.10** | **50.08** | **46.75** | **89.20** | **66.20** | **65.60** | **65.00** | **64.60** |
| | +MART | 63.68 | 38.97 | 42.80 | 42.77 | 37.62 | 61.80 | 36.40 | 41.80 | 41.60 | 35.40 |
| | +Ours | **74.89** | **47.60** | **51.18** | **51.16** | **45.97** | **88.00** | **65.00** | **64.40** | **64.40** | **63.40** |
| Swin-Ti | TRADES | 79.41 | 46.45 | 49.3 | 49.23 | 45.74 | 93.60 | 73.80 | 73.40 | 73.00 | 72.00 |
| | +Ours | **80.71** | **47.11** | **49.79** | **49.74** | **46.36** | **94.60** | **75.60** | **74.20** | **74.20** | **73.40** |
| | +MART | 75.19 | 46.10 | 49.82 | 49.71 | 44.54 | 92.40 | 71.60 | 70.20 | 69.60 | 68.60 |
| | +Ours | **77.37** | **46.98** | **50.44** | **50.28** | **45.28** | **96.20** | **80.00** | **70.80** | **70.60** | **70.00** |

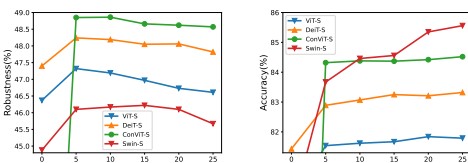
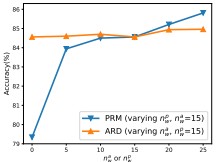

(a) Sensitivity of $n_w$ on overall approach.
(*Left*: AA Robustness, *Right*: Accuracy)

(b) Sensitivity of $n_w$ on each method individually.
(*Left*: AA Robustness, *Right*: Accuracy)

Figure 5: Performance of ViTs under different hyperparameter configurations.

best one in the last experiment), we change another hyperparameter (*e.g.*, $n_w^p$) as $\{0, 5, 10, 15, 20, 25\}$. In Figure 5 (b), for robustness, these two situations show different trends, *i.e.*, a longer warm-up with ARD (the orange line) increases the final robustness and a longer warm-up with PRM (the blue line) decrease the final robustness. We can find a good balance when $n_w^a = n_w^p = 15$. The former has improved the performance enough, and the latter has not decreased too much. As a result, we set $n_w^a = n_w^p = n_w$ by default.

# 5    Conclusion

In this paper, we conducted comprehensive experiments to investigate various training techniques to bring the first implementation benchmark for adversarial training of ViTs. We find that, under adversarial training, pre-training and gradient clipping are necessary while SGD is preferred over AdamW. Besides, taking the architectural information into consideration, we incorporated two simple but effective methods, ARD and PRM, into adversarial training to improve the adversarial robustness further. Extensive experiments demonstrated the effectiveness of our proposed methods. We hope our work not only provides the first implementation benchmark for adversarial training of ViTs, but also reminds researchers of the potential of architectural information contained within newly designed models like ViTs.

# Acknowledgment

Yisen Wang is partially supported by the NSF China (No. 62006153), Project 2020BD006 supported by PKU-Baidu Fund, and Open Research Projects of Zhejiang Lab (No. 2022RC0AB05).

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
