## A  Broader Impact

As brand-new models, the vulnerability of ViTs to adversarial samples motivates us to upgrade their security. Therefore, in this paper, we improve the adversarial robustness of ViTs in terms of strategies and architectures. Our approaches may contribute to a safer use of ViTs in the real world. However, the results also show that we are still far from fully robust ViTs. Meanwhile, our additional sampling process may bring negative impacts on the environmental protection (*e.g.* the emission of carbon dioxide).

## B  Gradient Clipping on CNNs

Table 6: The performance of adversarially trained ResNet50 and WideResNet50 with or without gradient clipping (GC).

| Model | CIFAR-10 | | Imagenette | |
|---|---|---|---|---|
| | Natural | AA | Natural | AA |
| ResNet50 | 82.47 | 49.02 | 92.00 | 61.00 |
| ResNet50+GC | **83.14** | **49.19** | **93.60** | 61.00 |
| WRN-50-2 | **86.09** | **51.33** | 92.60 | **62.00** |
| WRN-50-2+GC | 85.98 | 51.05 | **93.20** | 60.80 |

We have shown the necessity of gradient clipping (GC) for ViTs in Section 3.2. Here, we explore whether GC is beneficial to CNNs. We pretrain CNNs on ImageNet-1K. On CIFAR-10, we substitute the first convolutional layer (kernel size 7, stride 2, padding 3) with a convolutional layer without down-sampling (kernel size 3, stride 1, padding 1) similar to [16, 17, 15]. Finally, we adversarially train CNNs using the same settings as Section 3.2. The results in Table 6 demonstrate that gradient clipping has almost no effect on the robustness of CNNs. This indicates the necessity of gradient clipping is specific for ViTs.

## C  Adversarial Training Algorithm after Combining ARD and PRM

---
**Algorithm 1** Adversarial Training with ARD and PRM
---
1: **Input:** Network $f_\theta$, training data $\{(x_i, y_i)\}_{i=1}^n$, batch size $m$, learning rate $\eta$, PGD steps $N$, warming-up epoch $n_w$, number of splitted patches $J$, sampled variable for i-th attention block $u_i$, number of batches $R$, epoch $T$.
2: **Output:** Robust model $f_\theta$.
3: **for** $t = 0$ to $T - 1$ **do**
4:    **for** $a = 0$ to $R - 1$ **do**
5:        $p = 1 - \min(\frac{t}{n_w} + \frac{a+1}{Rn_w}, 1)$
6:        $k = p$
7:        $u_i \sim \text{Bernoulli}(p)$
8:        **for** $i = 1, \ldots, m$ (in parallel) **do**
9:            $\delta \sim \text{Uniform}(-\epsilon, \epsilon)$
10:           **for** $j = 1$ to $N$ **do**
11:               Randomly choose $\lfloor Jk \rfloor$ patches from all patches
12:               Generate mask $M$
13:               Update $\delta$ using Equation 9
14:           **end for**
15:           $x_i' \leftarrow x_i + \delta$
16:       **end for**
17:       $\theta \leftarrow \theta - \eta \nabla_\theta \frac{1}{m} \sum_i \mathcal{L}(f_\theta(x_i'), y_i)$
18:   **end for**
19: **end for**
---

# D  Evaluation on ImageNet

Table 7: The performance (%) of our overall algorithm on ImageNet-1K dataset.

| Model | Method | Natural | CW20 | PGD-20 | PGD-100 | AA |
|-------|--------|---------|------|--------|---------|-----|
| ViT-B | vanilla | 62.34 | 32.02 | 33.18 | 32.93 | 28.81 |
|       | +both | **69.10** | **38.92** | **37.96** | **37.52** | **34.62** |
| Swin-B | vanilla | 73.33 | 42.31 | 40.72 | 40.30 | 37.91 |
|        | +both | **74.36** | **43.16** | **41.37** | **40.87** | **38.61** |

In this section, we evaluate the proposed method on ImageNet-1K, the most commonly used large-scale dataset. We apply the most popular threat model on ImageNet-1K, *i.e.*, setting the perturbation budget $\epsilon = 4/255$. During adversarial training, ViTs are adversarially trained for only 10 epochs to save experimental time, and the learning rate is divided by 10 at the 6-th and 8-th epoch. We use PGD-5 with the step size $2/255$ to craft adversarial examples on the fly during training.

In Table 7, our method improves both natural accuracy and robustness by notable margins. For example, we improve the natural accuracy of ViT-B by 6.76%, and robustness (evaluated by AutoAttack) by 5.81%. Similarly, on Swin-B, we achieve robustness of 38.61%, which surpasses the best results on ImageNet-1K [32], which indicates that we benchmark a new state-of-the-art robustness on ImageNet-1K.