# OpenReview forum: "When Adversarial Training Meets Vision Transformers: Recipes from Training to Architecture"
_NeurIPS.cc/2022/Conference — NeurIPS 2022 Accept_

### Official Review · Reviewer_JwsQ · 2022-07-10

**Rating:** 5
**Confidence:** 4
**Soundness:** 2 fair
**Presentation:** 3 good
**Contribution:** 2 fair

**Summary:**

This paper presents two empirical strategies to improve the effectiveness of adversarial training in vision transformer models. The authors first analyzed the effects of gradient clipping, pre-training, data augmentations, optimizer, and training schedules. They further proposed two tricks, attention random dropping (ARD) and perturbation random masking (PRM) to improve the adversarial robustness. Evaluations show that both methods could contribute to the robustness enhancements.

**Questions:**

1. The authors do not deliver a further in-depth analysis or ablation study on the reason behind these two tricks.
2. In Table 3, the combination of ARD and PRM fails to deliver better results. I assume the reason could be the regularization will be too strong, but I expect to see the answers from the authors.

**Limitations:**

There is no potential negative societal impact of this work.

**Strengths And Weaknesses:**

Strengths:
1. The paper is well written, well-organized, and easy to follow.
2. The evaluation is thorough and sufficient.

Weakness:
1. There is no in-depth analysis or ablations on how these different vectors affect the adversarial training.
2. The technical novelty is limited, and the authors fail to provide intuition on why these two tricks would work.
3. Although the authors provide the chain rule of gradient propagation for these two tricks, the overall idea is still similar to dropout-style regularization. Therefore, the authors need to provide more baselines on different regularization methods.

---

> ### Author Response · Authors · 2022-08-02
> **Response to Reviewer JwsQ**
>
> Dear reviewer JwsQ:
>
> Thank you for your approval of the presentation and evaluation for this paper. Our responses to your proposed weakness and questions are as follows:
>
> **Q1:** There is no in-depth analysis or ablations on how these two methods (ARD and PRM) affect the adversarial training.
>
> **A1:** In fact, we did an ablation study in Section 4.6. Specifically, we illustrate how the performance changes when the hyperparameter varies for one method ($n_w^a \in [0, 25]$ for ARD or $n_w^p \in [0, 25]$ for PRM) in Figure 5(b), while the other is fixed ($n_w^p=15$ for PRM or $n_w^a=15$ for ARD respectively).
>
> **Q2:** The technical novelty is limited, and the authors don't provide intuition on why the two proposed tricks would work.
>
> **A2:** Our method is inspired by *curriculum learning*, which trains models from easier data to harder data. Curriculum learning-based methods [1, 2, 3] have been proven effective in improving adversarial robustness. Unlike these previous studies which generally increase the steps for adversarial example generation during training, we randomly mask some gradient information (through architecture and input) during generation to craft easier adversarial examples at the first several training epochs.
>
> **Q3:** The idea in this paper is similar to dropout-style regularization. The authors should provide more comparisons with our regularization methods.
>
> First, we want to clarify the difference between the proposed method and dropout-style regularization. We only apply the proposed method to adversarial example generation. Once the adversarial examples are crafted as training data, gradient calculation (including forward and backward propagation) for parameter update is the same as normal training. By contrast, common dropout-style regularization always masks some components when calculating gradients used to update parameters.
>
> In addition, we compare our method with other commonly-used dropout regularization including dropout [4], attention dropout [5], and stochastic depth [6]. They all improve the performance in the natural training of transformers. We follow their settings and evaluate them on adversarial training of DeiT-S on CIFAR-10. As shown in the table below, except for our method, none of them improve the test performance.
>
> |                    | Natural   | PGD-20    | AA        |
> | ------------------ | --------- | --------- | --------- |
> | Vaniila            | 81.43     | 51.88     | 47.40     |
> | +Dropout           | 75.23     | 46.74     | 42.95     |
> | +Attention Dropout | 79.83     | 50.61     | 46.20     |
> | +Stochastic depth  | 80.51     | 51.09     | 46.93     |
> | +Ours (ARD \& PRM) | **83.07** | **52.26** | **48.19** |
>
> **Q4:** Can the authors explain why the combination of ARD and PRM fails to deliver better results?
>
> **A4:** In Table 3, we always fix the hyperparameter $n_w=10$ in all experiments, which may ignore the interaction between these two approaches and make the effect too strong. To reveal the potential of the combined method, we use fewer epochs for warmup with our proposed method when we adversarially train ViT-S on CIFAR-10. In the table below, the combined method (ARD + PRM) with $n_w=5$ outperforms the individual method (ARD or PRM) with $n_w=10$.
>
> |         | vanilla | +ARD  | +PRM  | +both ($n_w$=10) | +both ($n_w$=5) |
> | ------- | ------- | ----- | ----- | ---------------- | --------------- |
> | ROB(\%) | 46.37   | 47.12 | 47.22 | 47.19            | 47.32           |
>
> We sincerely hope that our response could answer your questions and you could reconsider your rating. Looking forward to the further discussion with you in the Author-Reviewer Discussion period.
>
> **References:**
>
> $[1]$ Yisen Wang, Xingjun Ma, James Bailey, Jinfeng Yi, Bowen Zhou, and Quanquan Gu. On the Convergence and Robustness of Adversarial Training. In ICML, 2019.
>
> $[2]$ ingfeng Zhang, Xilie Xu, Bo Han, Gang Niu, Lizhen Cui, Masashi Sugiyama, and Mohan Kankanhalli. Attacks Which Do Not Kill Training Make Adversarial Learning Stronger. In ICML, 2020.
>
> $[3]$ Chawin Sitawarin, Supriyo Chakraborty, and David Wagner. SAT: Improving Adversarial Training via Curriculum-Based Loss Smoothing. In AISec, 2021.
>
> $[4]$ Alexey Dosovitskiy, Lucas Beyer, Alexander Kolesnikov, Dirk Weissenborn, Xiaohua Zhai, Thomas Unterthiner, Mostafa Dehghani, Matthias Minderer, Georg Heigold, Sylvain Gelly, Jakob Uszkoreit, and Neil Houlsby. An image is worth 16x16 words: Transformers for image recognition at scale. In ICLR, 2021.
>
> $[5]$  Ara Abigail E. Ambita, Eujene Nikka V. Boquio, and Prospero C. Naval Jr. COViT-GAN: Vision Transformer for COVID-19 Detection in CT Scan Images with Self-Attention GAN for Data Augmentation. In ICANN, 2021.
>
> $[6]$ Hugo Touvron, Matthieu Cord, Matthijs Douze, Francisco Massa, Alexandre Sablayrolles, and Herve Jegou. Training data-efficient image transformers & distillation through attention. In ICML, 2021.

---

> > ### Comment · Reviewer_JwsQ · 2022-08-06
> > **Ack to Author**
> >
> > I thank the authors for providing new results and clarification. However, these two tricks are still like some black magic to me. Given the results, I would slightly raise my rating to 5.

---

> > > ### Author Response · Authors · 2022-08-08
> > > **Further Response to Reviewer JwsQ**
> > >
> > > Thank you for the reply. We would like to further explain the intuitive motivation behind the proposed method. According to *Curriculum Learning*, if training data are too hard at the beginning, the model is unable to learn effectively. Since the information of the model is fully used during generation, the generated adversarial examples may be very strong, leading to difficulties in the initial stage of adversarial training. To mitigate this problem, we remove some information (through patches and the architecture) to craft easier adversarial examples and help the model to learn effectively. Specifically, we remove gradients that flow through some multi-head attention modules for ARD, and also remove the gradients on some patches for PRM. As the training proceeds, less and less information is removed, and the generation uses all the information again after $n_w$ epochs. The proposed method currently is a little bit heuristic, but it indeed brings notable improvements on the robustness of ViTs. We believe that this is only a start point and we will keep pursuing its detailed analysis and theoretical support in the future.

---

> ### Author Response · Authors · 2022-08-06
> **Need further clarification?**
>
> Thanks very much for your constructive and detailed comments. We have tried our best to address the concerns and revised our paper accordingly. Is there any unclear point that we should/could further clarify?

---

### Official Review · Reviewer_nQpy · 2022-07-11

**Rating:** 9
**Confidence:** 5
**Soundness:** 4 excellent
**Presentation:** 4 excellent
**Contribution:** 4 excellent

**Summary:**


Summary:

I think this article makes a huge contribution to the adversarial robustness of ViT. It is not just a simple application of adversarial training suitable for CNN to ViT, but it has made substantial contributions, including detailed and reasonable analysis and proposing new AT techniques suitable for ViT. I like this paper very much and look forward to its early publication. It would undoubtedly be a great contribution to the community if the authors could publish the trained robust models in time. I'm looking forward to it being included in the robustness library.


**Questions:**



Please answer the questions in "Strengths And Weaknesses."



**Limitations:**


The authors adequately addressed the limitations and potential negative societal impact of their work.



**Strengths And Weaknesses:**


(Positive) This paper provides a comprehensive evaluation of various training techniques across several datasets, thus bringing the first implementation benchmark for adversarial training of ViTs.


(Positive) It is not just a simple application of adversarial training suitable for CNN to ViT, but it has made substantial contributions, including detailed and reasonable analysis and proposing new AT techniques suitable for ViT.

(Positive) CNN and ViT are indeed very different, so the AT of CNN should be different from that of ViT. For example, compared to CNN, ViT requires Gradient Clipping and pre-training, prefers advanced DA, and is more prone to robust overfitting.


(Positive) Although we know the difference between CNN and ViT under vanilla training, relying on this known knowledge is not enough. For example, we know that ViT favors AdamW in Vanilla training, but the authors found that adversarial training of ViT favors SGD.

(Positive) The authors put forward two distinctive AT methods, PMR and ARD, specifically for the characteristics of ViT.

(Positive) Indeed, as the authors say, although ViT and CNN are very different, ViT trained only by vanilla training is not significantly robust (only a small effect under extremely small interference).

(Positive) According to the protocol on CIFAR-10, it is reasonable for the authors to change the input of ViT to 32x32.

(Positive) Similarly, in my experience, it is reasonable for the authors to use AA to test the robustness of ViT.

(Positive) It is reasonable for the authors to use PGD-10 to perform AT on ViT.


(Positive) The authors' results on the AT for ViT are reasonable and hopefully will be published sooner for people to use.


(Negative) Figure 2 shows that the effect of pre-training with ImageNet-1k is slightly higher than that of ImageNet-21k. Can the authors give some reasonable explanations?



(Negative) The content of formula (4) to formula (7) is too simple, and the existence of these formulas is unnecessary. These formulas can be simplified or omitted unless the authors want to use these formulas to give theoretical results to their method.


(Question) It can be seen from Table 3 that the effect of PRM is better than that of ARD in general. I hope the authors give some reasonable explanations.

(Positive) The authors demonstrate that their method can be combined with and improve the performance of existing AT methods. This is very inspiring.

(Positive) The authors are commendable for their hyperparameter analysis of the number of epochs for warmup.

---

> ### Author Response · Authors · 2022-08-02
> **Response to Reviewer nQpy**
>
> Dear reviewer nQpy:
>
> Thank you very much for your approval and recognition for this paper. Like you said, in this paper, we not only provide the first benchmark for AT of ViTs, but also propose new AT methods for ViTs. Our contribution may help the community better understand ViTs in the adversarial region and open a line of works for improving adversarial robustness of ViTs in the future. We will release our code and pretrained models immediately upon acceptance. For your concerns, we list our responses as follows:
>
> **Q1:** In Figure 2, why is the effect of pre-training with ImageNet-1k is slightly higher than that of ImageNet-21k?
>
> **A1:** That is an insightful observation, since it is unusual that ImageNet-21k pretraining underperforms ImageNet-1k pretraining. We double checked our code and ensured that there are no bugs. Here, we provide some preliminary insights on it.
>
> In our experiments, the two pretraining strategies have similar results on CIFAR-10, while the difference is more clear on ImageNette, a 10-class subset of ImageNet-1k. This fact leads us to conjecture that the **closer** relationship between ImageNet-1k and ImageNette could benefit feature transferability a lot. ImageNet-21k, on the other hand, contain 21x more classes than ImageNet-1k, so its link to ImageNette could be much weaker. This observation could help explain why ImageNet-21k performs worse than ImageNet-1k. Nevertheless, since adversarial training is quite different from standard training, transferring from standard training to adversarial training might behave some unexpected phenomena. We leave a more thorough investigation of this phenomenon to future work.
>
> **Q2:** Considering Eqn. (4) and Eqn. (7) are too simple, they could be simplified or omitted.
>
> **A2:** We want our paper to be as easy-to-understand as possible for readers from different grounds, especially for those who are not familiar with model architectures. So we provide concrete formulas like Eqn. (4) and Eqn. (7). We will simplify them appropriately in the revision.
>
> **Q3:** Why does PRM perform better than ARD in general as shown in Table 3?
>
> **A3:** In our opinion, PRM is a more aggressive approach in its mechanism, because ARD only removes part of intermediate gradient information during crafting adversarial samples (this is because, even though we block gradient to flow through some blocks, the parallel skip connections always allow the gradient to flow from the higher layers to the lower layers) while PRM may remove most of the adversarial perturbations on the image patches, especially at the early stage of training. In our experiments, we only use a shared hyperparameter $n_w$ for both ARD and PRM for simplicity. We find that $n_w$ affects the performance of both methods in different ways in Figure 5(b). ARD increases its performance with larger $n_w$, while PRM decreases its performance with too larger $n_w$. Thus, we set $n_w=10$ to trade off their performance, which results in a relatively weak performance of ARD. We can further improve the performance of ARD by a more careful selection of the independent hyperparameter $n_w^a$ for ARD rather than a shared hyperparameter.
>
> Thank you again for your positive comments on this paper. Looking forward to the further discussion with you in the Author-Reviewer Discussion period.

---

> ### Author Response · Authors · 2022-08-06
> **Need further clarification?**
>
> Thanks very much for your constructive and detailed comments. We have tried our best to address the concerns and revised our paper accordingly. Is there any unclear point that we should/could further clarify?

---

> ### Comment · Reviewer_nQpy · 2022-08-08
> **post-rebuttal comment**
>
>
> Many thanks to the authors for their response; my concerns were appropriately addressed. Also, I read the comments of other reviewers. Thanks to all the reviewers for their insightful comments and the authors for their detailed responses.
>
> Best regards,

---

### Official Review · Reviewer_63WY · 2022-07-13

**Rating:** 5
**Confidence:** 5
**Soundness:** 3 good
**Presentation:** 3 good
**Contribution:** 3 good

**Summary:**

This paper analyzes and improves the robust training performance with vision transformers. If first rigorously analyze the impact of various factors on robust training and then proposes novel methods, namely attention random dropping (ARD) and perturbation random masking (PRM), to further boost the performance.

**Questions:**

I am also puzzled why a very small number of epochs are chosen (40 by default). ViTs are generally trained for longer schedules (often 300 epochs), especially when trained from scratch. Was computational cost the only reason behind this design choice?

Similarly, another puzzling trend is in fig 2. In fig. 2a, pre-training on ImageNet-1k achieves better performance than pertaining on much bigger ImageNet-21K. This is strange as ImageNet-21K tends to always perform better. Similarly in fig. 2b, the training accuracy of the model itself doesn’t converge to very high values, suggesting that the bottleneck in pre-training from scratch is the optimization process, not generalizing. Given the large number of parameters in ViTs, shouldn’t they be able to fit the data on the training set?

**Limitations:**

The limited performance of ViTs, in comparison to CNNs, is not highlighted in the paper. I encourage authors to discuss this limitation with concrete comparisons.

**Strengths And Weaknesses:**

This paper is among the first works to analyze robust training performance on vision transformers. It uniquely takes a deep dive into the fields by rigorously analyzing the impact of various factors in robust training on the final performance.

----

The major limitation of this work is the highly sub-optimal performance of vision transformers in adversarial training. For example, after all proposed innovations (including pre-training, warmup strategies, careful augmentations, etc.) the best robust accuracy (AA) achieved on cifar-10 is 50.01 with ViT-B. In contrast, a ResNet-18 achieves 51.06 robust accuracy (AA) without any pre-training (so no extra data or other bags of tricks) [Source: RobustBench]. Note that a ResNet-18 has at least 10x (if not more) lower number of parameters than ViT-B. With extra data, the ResNet-18 network can further achieve 57.67 robust accuracy (AA), much better than the best achieved with very large ViTs in the paper.

The aforementioned critique also signals towards another critical comparison that is missing in the paper: A comparison of ViTs and CNNs performance. ViTs continued to be adopted in the community over CNN since they outperformance CNNs. To make a case for the use of ViTs in a robust training regime, one needs to show a similar benefit of ViTs over CNNs. However, I couldn’t find any such comparison in the paper (as I argued above the performance of ViTs is far suboptimal than CNNs in the current paper).

Another troubling trend on the performance front (cifar-10) is that Swin, which is much better architecture than baseline ViT-B, has poorer performance (table 1, 2). Not just for Swin, but this phenomenon is also true for other modern architectures that are better than baseline ViT. This makes me question whether the proposed innovations are any effective at lower resolutions since they are unable to utilize better architectures. Only at higher resolution (ImageNette), they perform better with better architectures. This should be clearly clarified in the paper.

---

> ### Author Response · Authors · 2022-08-02
> **Response to Reviewer 63WY (1/2)**
>
> Dear reviewer 63WY:
>
> We appreciate your valuable comments and suggestions, which enable us to deeply consider the research significance of ViT's adversarial robustness.
>
> **Q1:** ViTs in this paper only achieve highly sub-optimal performance (51.01% by ViT-B) on CIFAR-10 compared to CNNs (51.06% by PreActResNet-18).
>
> **A1:** Considering this paper is among the first works to analyze robust training performance on vision transformers, we argue that we should not be overly pessimistic about the performance of adversarial training for ViTs. Especially, CIFAR-10 are low-resolution datasets, while ViTs are usually pretrained on high-resolution datasets. We might need a better adaptation method for this input size difference.
>
> On high-resolution datasets like ImageNette and ImageNet-1K, the robustness results are totally different as shown in tables below. On Imagenette, transformers of different sizes (Swin-Ti for small size, and Swin-S for medium size) outperform their competitors (ResNet50 or WideResNet-50-2 respectively) with a similar even smaller number of parameters by a notable margin (>10%). On ImageNet-1K (a more difficult dataset for adversarial tasks), Swin-B with more parameters also obtains almost 4% robustness improvements compared to ResNet50.
>
> Therefore, we think ViTs have unrealized potential on CIFAR-10. Revisiting the development history of adversarial training, at the very beginning WideResNet-34-10 with original experimental settings only achieved 44.04% of robustness (See \#61 in RobustBench) on CIFAR-10 in 2018, while its robustness increased rapidly in the following years. We believe that, if we can provide a solid benchmarking result first on ViTs, its adversarial robustness on CIFAR-10 will also be likely to increase rapidly via emerging studies.
>
> The performance of different CNNs and ViTs on ImageNette.
>
> | Models  | ResNet50 | Swin-Ti | WRN-50-2 | Swin-S |
> | ------- | -------- | ------- | -------- | ------ |
> | #Params | 22.5M    | 27.5M   | 66.9M    | 48.8M  |
> | ROB (%) | 56.33    | 67.34   | 57.71    | 68.56  |
>
> The performance of different CNNs and ViTs on ImageNet-1K.
>
> | Models  | ResNet50 | Swin-B |
> | ------- | -------- | ------ |
> | #Params | 22.5M    | 86.8M  |
> | ROB (%) | 34.96    | 38.61  |
>
> **Q2:** This paper lacks a comparison between ViTs and CNNs, and never shows the benefits of ViTs over CNNs in terms of adversarial training.
>
> **A2:** Except the results on low-resolution dataset CIFAR, we list more results on high-resolution datasets like ImageNette and ImageNet-1K in below tables to compare the performance of ViTs and CNNs.
>
>  The performance of different CNNs and ViTs on ImageNette.
>
> | Models  | ResNet50 | Swin-Ti | WRN-50-2 | Swin-S |
> | ------- | -------- | ------- | -------- | ------ |
> | #Params | 22.5M    | 27.5M   | 66.9M    | 48.8M  |
> | ROB (%) | 56.33    | 67.34   | 57.71    | 68.56  |
>
> The performance of different CNNs and ViTs on ImageNet-1K.
>
> | Models  | ResNet50 | Swin-B |
> | ------- | -------- | ------ |
> | #Params | 22.5M    | 86.8M  |
> | ROB (%) | 34.96    | 38.61  |
>
> We find that ViTs have already achieved higher robustness on high-resolution datasets (e.g., Imagenette, ImageNet-1K). On Imagenette, transformers of different sizes (Swin-Ti for small size, and Swin-S for medium size) outperform their competitors (ResNet50 or WideResNet-50-2 respectively) with a similar even smaller number of parameters by a notable margin (>10%). On ImageNet-1K (a more difficult dataset for adversarial tasks), Swin-B with more parameters also obtains almost 4% robustness improvements compared to ResNet50. The temporary sub-optimal performance on low-resolution datasets (e.g., CIFAR-10), in our opinion, is due to the currently inadequate research on adversarial training of ViTs (See Q1\&A1), which would be mitigated as long as a solid benchmark result is provided first.

---

> > ### Author Response · Authors · 2022-08-02
> > **Response to Reviewer 63WY (2/2)**
> >
> > **Q3:** Why does Swin, a better architecture on natural recognition, perform worse than ViT on robustness?
> >
> > **A3:** Swin performs poorly on CIFAR-10 (a low-resolution dataset) in Table 1 and Table 2, while still performing well on Imagenette and ImageNet (high-resolution datasets) in Table 1, Table 2, and Table 7. We conjecture, the inductive bias (multi-scale objects and the high resolution of pixels on high-resolution images) inside Swin is no longer valid on low-resolution images, which results in unsatisfactory performance. In either case, our method shows notable and consistent improvements compared to the baseline, which proves the effectiveness of the proposed method. We will clarify it in our revision.
> >
> > **Q4:** Why do we use only 40 epochs instead of more epochs for adversarial training of ViTs?
> >
> > **A4:** We have already tried to train ViT for more epochs in Section 3.5, and the test performance instead decreased even though its train performance increases. The training curve can be found in Figure 3(b), and we also show the performance of the last checkpoints in the table below.
> >
> > | Epochs | Train ACC | Train ROB | Test ACC | Test ROB |
> > | ------ | --------- | --------- | -------- | -------- |
> > | 40     | 93.43     | 68.35     | 85.53    | 50.76    |
> > | 80     | 97.05     | 77.93     | 83.87    | 46.27    |
> >
> > This phenomenon is termed *robust overfitting* [1], i.e., the robustness in the test set will only continue to substantially decrease with further training after a certain point in adversarial training. In conclusion, we use only 40 epochs to mainly avoid robust overfitting, in addition to its less computational cost.
> >
> > **Q5:** Why do ViTs pretrained on ImageNet-21K (a larger dataset) perform worse than ViTs pretrained on ImageNet-1K?
> >
> > A5: That is an insightful observation, since it is unusual that ImageNet-21k pretraining underperforms ImageNet-1k pretraining. We double checked our code and ensured that there are no bugs. Here, we provide some preliminary insights on it.
> >
> > In our experiments, the two pretraining strategies have similar results on CIFAR-10, while the difference is more clear on ImageNette, a 10-class subset of ImageNet-1k. This fact leads us to conjecture that the **closer** relationship between ImageNet-1k and ImageNette could benefit feature transferability a lot. ImageNet-21k, on the other hand, contain 21x more classes than ImageNet-1k, so its link to ImageNette could be much weaker. This observation could help explain why ImageNet-21k performs worse than ImageNet-1k. Nevertheless, since adversarial training is quite different from standard training, transferring from standard training to adversarial training might behave some unexpected phenomena. We leave a more thorough investigation of this phenomenon to future work.
> >
> > **Q6:** Why do ViTs trained from scratch fail to fit the data even with a large number of parameters?
> >
> > **A6:** In fact, ViTs trained from scratch get stuck in poor local minima. As indicated by a previous study [2], ViT has a more non-convex loss landscape, which is verified by the fact that ViT has more negative eigenvalues on loss landscape than CNNs. In our case, enormous training data on the pretraining dataset help ViT escape these poor local minima, and such pretrained parameters provide a good initialization for ViTs on small datasets. By contrast, a randomly initialized ViT easily falls in poor local minimum after being directly trained on a small dataset.
> >
> > We really hope our explanation can eliminate your doubts. We are happy for the further discussion with you in the following Author-Reviewer Discussion period.
> >
> > **References:**
> >
> > $[1]$ Leslie Rice, Eric Wong, and J. Zico Kolter. Overfitting in adversarially robust deep learning. In ICML, 2020.
> >
> > $[2]$ Namuk Park and Songkuk Kim. How Do Vision Transformers Work?  In ICLR, 2022.

---

> > > ### Comment · Reviewer_63WY · 2022-08-09
> > > **Unexplained trends in experiments**
> > >
> > > I thanks authors for addressing some of the concerns (e.g., comparison of cnns and vits). However, as authors discussed, some the key trends in the experiments remain unexplained. I still believe that the paper can be significantly improved by addressing these limitations.
> > >
> > > E.g., regarding imagenette, its crucial not to use the v2 [1] of the dataset since its validation data includes training images from original ImageNet. When using Im1K pretraining, then we are directly learning from validation data for imagenette, which violates the most basic principle in machine learning.
> > >
> > > Similarly, its remain unclear why better architectures are performing worse in adversarial training (Swin performs much worse). In parallel, there is extreme overfitting (only 40 epochs are used), hindering longer training, especially since ViTs require long training schedules to achieve high performance. Note that both of these *troubling trends* don't align with other complementary works, which, even with a straightforward adversarial training, don't observe any such trends [2, 3]. Overall I recommend double-checking the adversarial training implementation and ensure that these trends aren't arising from an bug in implementation.
> > >
> > > 1. https://github.com/fastai/imagenette
> > > 2. Shao, Rulin, et al. "On the adversarial robustness of vision transformers." arXiv preprint arXiv:2103.15670 (2021).
> > > 3.  Adversarially Robust Vision Transformers (https://edoardo.science/thesis.pdf)

---

> > > > ### Author Response · Authors · 2022-08-09
> > > > **Further explanation on experiments**
> > > >
> > > > **Q1:**  It is crucial not to use the v2 [1] of the dataset since its validation data includes training images from the original ImageNet-1K when using ImageNet-1K pretraining.
> > > >
> > > > **A1:** We thank the reviewer for pointing out this detail. After carefully checking the Imagenette dataset, we find there exist some Imagenette test data which appear on the ImageNet-1K training set. Thus, we constructed a test set after removing these overlapping data, letting models never see any test data during training and pre-training. The robustness evaluated by AutoAttack are shown in the tables below, and we find that the results are still similar to the original ones. This is reasonable because natural pre-training on large datasets does not improve adversarial robustness as indicated in previous studies [2, 4]. A set of the pretraining parameters in our paper are only for better initialization for the subsequent training. Further, we will update all experimental results on Imagenette in the revision and open-source the non-overlapping test set as a benchmark dataset for the adversarial robustness on ViTs for the following fair comparisons. In addition, the superior performance of our method on ImageNet dataset can still verify its effectiveness.
> > > >
> > > > On Imagenette
> > > > |         | DeiT-Ti  | DeiT-Ti | ConViT-Ti | ConViT-Ti | Swin-Ti  | Swin-Ti |
> > > > | ------- | -------- | ------- | --------- | --------- | -------- | ------- |
> > > > |         | Original | Removed | Original  | Removed   | Original | Removed |
> > > > | vanilla | 50.60    | 55.97   | 31.34     | 34.44     | 65.78    | 66.42   |
> > > > | +ARD    | 55.77    | 57.46   | 57.30     | 61.19     | 65.96    | 69.40   |
> > > > | +PRM    | 55.44    | 58.21   | 44.43     | 45.52     | 67.21    | 72.39   |
> > > > | +both   | 55.80    | 58.96   | 57.48     | 62.69     | 67.34    | 73.88   |
> > > >
> > > > On ImageNet
> > > > | Model  | Method   | AA    |
> > > > | ------ | ------- |   ----- |
> > > > | ViT-B  | vanilla | 28.81 |
> > > > | ViT-B  | +both   | 34.62 |
> > > > | Swin-B | vanilla | 37.91 |
> > > > | Swin-B | +both   | 38.61 |
> > > >
> > > >
> > > >
> > > > **Q2:** The trends discovered in this paper don't align with other complementary works.
> > > >
> > > > **A2:** First, we would like to clarify the difference between our paper and other complementary works [2, 3]. Our paper focuses on the adversarial robustness of various **adversarially trained models**, while previous research focuses more on the adversarial robustness of various **naturally trained models** (e.g., the previous study [2] only adversarially trained a single ViT-B in Table 3, while the other robust results are all about naturally trained models). Since adversarial training is quite different from natural training, different trends might be discovered. Specifically, adversarial training always suffers from severe robust overfitting after longer training, which may outweigh the benefits of the long training schedule to vision transformers.
> > > >
> > > > Second, in Table 2 of [2] on ImageNet, it is **naturally trained Swin**, which obtains worser robustness compared to naturally trained ViTs. While for **adversarial training**, Swin performs poorly on CIFAR-10 (a low-resolution dataset) in Table 1 and Table 2, while still performing better on ImageNet (high-resolution datasets) in Table 7.
> > > >
> > > > Finally, considering that this paper is one of the first works to investigate the adversarial training of vision transformers systematically, our mission is to explore some non-trivial and intriguing phenomena to inspire more future studies.
> > > >
> > > > **References:**
> > > >
> > > > [1] https://github.com/fastai/imagenette
> > > >
> > > > [2] Rulin Shao, Zhouxing Shi, Jinfeng Yi, Pin-Yu Chen and Cho-Jui Hsieh. On the adversarial robustness of vision transformers. In arXiv, 2021.
> > > >
> > > > [3] Adversarially Robust Vision Transformers (https://edoardo.science/thesis.pdf)
> > > >
> > > > [4] Dan Hendrycks, Kimin Lee, and Mantas Mazeika. Using pre-training can improve model robustness and uncertainty. In ICML, 2019.

---

> > > > > ### Comment · Reviewer_63WY · 2022-08-09
> > > > > **Concerns regarding correctness and competitiveness still remains**
> > > > >
> > > > > I thank the authors for looking into the issue with the imagenette dataset. Even though the trend doesn’t change I suggest that authors re-run experiments in Tables 1, 2, 3, and 4 in the main paper and Tables 5, and 6 in the Appendix to provide updated numbers.
> > > > >
> > > > > However, I still stand by my concerns regarding **correctness**, i.e., that experimental results of even the baselines are not competitive, likely due to a bug or poor implementation of adversarial training itself.
> > > > > - For example, a ConViT only achieves 29-34% and Swin only achieves 43-46% robust accuracy with robust training on cifar10. The poor performance of these powerful architecture raises the first red flag.
> > > > > - Even with proposed innovations, the best possible robust accuracy is 50.7% cifar10 (Table 3). In contrast, previous work has achieved 49.2 (Table 3 in [1]) with traditional adversarial training techniques (with an idential ViT-B architecture). This raises concerns about whether the true advantage of ARD and PRM is as high as claimed in experimental results.
> > > > >
> > > > > Overall. The poor performance of baselines adversarial training itself raises questions about the efficacy of ARD and PRM. While Table 3 gives the impression that ARD/PRM is highly effective, it might be simply because of poor baselines, i.e., the baseline numbers from ViTs adversarial training are not competitive. Since ARD/ARM are novel contributions, it is imperative to establish a strong baseline to validate the benefit of proposed techniques.
> > > > > - My suggestion is that authors carefully check the implementation of adversarial training since it will both 1) provide a strong baseline and 2) hopefully even improve the performance of ARD/ARM further.
> > > > >
> > > > > [1] Rulin Shao, Zhouxing Shi, Jinfeng Yi, Pin-Yu Chen and Cho-Jui Hsieh. On the adversarial robustness of vision transformers. In arXiv, 2021.

---

> > > > > > ### Author Response · Authors · 2022-08-09
> > > > > > **Clarification on the correctness**
> > > > > >
> > > > > > Dear Reviewer 63WY,
> > > > > >
> > > > > > Thanks for your reply! We will definitely update the results.
> > > > > >
> > > > > > We want to emphasize the results on CIFAR again. Please note this is **adversarial training**. As you have noted, **in Table 3 of [1], ViT-B achieved 49.2% robustness under AA, while in our Table 2, ViT-B achieved 49.06%**. This is **exactly the same baseline result**, which demonstrates the correctness of our running of baselines.
> > > > > >
> > > > > > For the effectiveness of ARD and PRM, we would like to remind you again here that this is **adversarial training**. As we all know, **in the adversarial training scenario, the improvement of robustness of about 1% is very difficult and significant**. As you have recognized, "with proposed innovations, the best possible robust accuracy is 50.7% cifar10", which has about 1% gain over baselines.
> > > > > >
> > > > > > Moreover, **it is not fair only focus on the results on the low-resolution dataset CIFAR, and we should also take the results on high-resolution datasets into consideration due to the fact that ViTs are usually pretrained on high-resolution datasets**. **As we have replied in previous Q1 & A1, on high-resolution datasets like ImageNette and ImageNet-1K, the robustness results are totally different**. On Imagenette, transformers of different sizes (Swin-Ti for small size, and Swin-S for medium size) outperform their competitors (ResNet50 or WideResNet-50-2 respectively) with a similar even smaller number of parameters by a notable margin (>10%). On ImageNet-1K (a more difficult dataset for adversarial tasks), Swin-B with more parameters also obtains almost 4% robustness improvements compared to ResNet50.
> > > > > >
> > > > > > Therefore, **we think ViTs have unrealized potential on CIFAR-10, as this paper is among the first works to analyze robust training performance on vision transformers**. Revisiting the **development history of adversarial training, at the very beginning WideResNet-34-10 with original experimental settings only achieved 44.04% of robustness (See #61 in RobustBench) on CIFAR-10 in 2018, while its robustness increased rapidly in the following years**. We believe that, if we can provide a solid benchmarking result first on ViTs, its adversarial robustness on CIFAR-10 will also be likely to increase rapidly via emerging studies.
> > > > > >
> > > > > > Lastly, we provide another piece of evidence. If we look into Table 2 of [1], the setting changed to naturally trained ViTs (NOT adversarial training), the attack success rate of Swin-S is 81.8% while ViT-S is 77.6%, i.e., Swin performs worse robustness than ViT against adversarial examples. However, from common sense, we would like to believe Swin should perform better than ViT, while this is actually the contrary. Therefore, we should remind ourselves that the conclusion under adversarial scenarios maybe different from natural settings.
> > > > > >
> > > > > > Hope our reply could address your concerns.

---

> ### Author Response · Authors · 2022-08-06
> **Need further clarification?**
>
> Thanks very much for your constructive and detailed comments. We have tried our best to address the concerns and revised our paper accordingly. Is there any unclear point that we should/could further clarify?

---

### Official Review · Reviewer_UDNi · 2022-07-16

**Rating:** 5
**Confidence:** 3
**Soundness:** 3 good
**Presentation:** 3 good
**Contribution:** 2 fair

**Summary:**

This paper benchmark tests several implementation details on adversarial training with Vision Transformers. Moreover, this paper proposes two modules, i.e., ARD and PRM to improve the adversarial robustness.

**Questions:**

In table 1, why ViT-B has the exact same awful performance for natural images and adversarial examples before the gradient clip? Does that mean that ViT-B does not work on natural image classification at all? And about the accuracy on Cifar-10 dataset, the 10.00% seems pretty magical. Do you explore whether this number is because the adversarial examples from nine out of ten classes can fool the model 100 percent, while the remaining one can not fool the model at all, or the number 10.00% just magically happens?

**Limitations:**

While there are lots of benchmark tests that provide empirical hints for practical applications, I personally expect discussions about the reason for such observations. For example, why ViT-B suffers the gradient explosion while Swim-B does not? How about other Transformer models? Why “SGD+piecewise” performs better than “AdamW+cyclic”, and is that true for all Transformer models?
With such discussions, I think it would be more useful in guiding practical applications.

Moreover. some conclusions are very difficult to be applied in practice. For example, the authors claim that “it is a good choice to use advanced data augmentations (e.g., Cutmix and Mixup), …”, but “RandAugment is a more advanced technique, which is too difficult for AT of ViTs”, then how “advanced” the data augmentation should be used?

The proposed ARD and PRM modules look interesting, but their effectiveness is not well justified by the experiments, especially combining them together shows lower performance in many cases. Is there any reason for this phenomenon?


**Strengths And Weaknesses:**

There are a number of benchmark tests, which is really appreciated.
The proposed ARD and PRM modules look interesting.

---

> ### Author Response · Authors · 2022-08-02
> **Response to Reviewer UDNi**
>
> Dear reviewer UDNi:
>
> We really appreciate your positive comments on this paper and our responses for your questions and limitations are as follows:
>
> **Q1:** Does ViT-B fail in working on natural image classification without gradient clipping?
>
> **A1:** Actually, we only evaluate the adversarially trained model on natural examples and adversarial examples in Table 1, which only indicates the failure of ViT-B on adversarial training **without** gradient clipping. In terms of ViT on natural training, after conducting extra experiments with the same settings as adversarial training, we find naturally trained ViT-B achieves 97.96% (10.00%) accuracy with (without) gradient clipping. This indicates ViT-B indeed works on natural image classification, although its success heavily relies on gradient clipping.
>
> **Q2:** Why does ViT-B achieve exactly 10.00% accuracy on both natural images and adversarial samples from CIFAR-10?
>
> **A2:** Without gradient clipping, ViT-B collapses during training and always predicts ‘’Airplane‘’ (class 0) on any inputs. Considering the sample numbers of different classes on CIFAR-10 test set are the same, the accuracy for the collapsed model is 10.00%.
>
> **Q3:** Why does ViT-B suffer from gradient explosion while Swin-B does not? How about other Transformer models?
>
> **A3:** The ViT-B is the standard Transformers from NLP with the fewest possible modifications on vision tasks, which may not be very adaptable to vision tasks (e.g., suffering from gradient explosion). DeiT-S and ConViT-B still suffer from gradient explosion since only limited modification occurs on both models compared to the vanilla ViT. By contrast, Swin-B applies more inductive biases in vision tasks (e.g., multi-scale and shift invariance) into the architecture, which smooth the loss landscape and mitigates the gradient explosion problem [1].
>
> **Q4:** Why does "SGD+piecewise'' perform better than "AdamW+cyclic''? Is it true for all Vision Transformer models?
>
> **A4:** Adversarial training suffers from severe robust overfitting, i.e., the model quickly overfits training data with a small learning rate, which leads to the decrease in the test performance. The piecewise schedule keeps the large learning rate at the first 35 epochs (40 in total), which prevents model from quickly overfitting training data. Besides, unlike AdamW, SGD does not normalize the gradient, which may make the parameter updates with more noise and thus avoid overfitting. To verify if  ‘’SGD+piecewise‘’ still works better  for other transformers, we conduct an extra experiment in the table below, and find DeiT-S also works better with ‘’SGD+piecewise‘’.
>
> |               | Natural   | AA        |
> | ------------- | --------- | --------- |
> | AdamW+cyclic  | 81.39     | 46.87     |
> | SGD+piecewise | 81.43 | 47.40 |
>
> **Q5:** How should the “advanced” data augmentation be used to improve the adversarial robustness of ViTs?
>
> **A5:** Since adversarial training suffers from severe robust overfitting [2], it will be better to use some reasonably strong data augmentation (e.g., Cutmix and Mixup) to reduce the gap between the test and training performance. However, if the ‘’advanced‘’ data augmentation is too strong (making the training too hard), it decreases the training performance by a notable margin, resulting in a lower test performance even with a small gap. For example, adversarial training with RandAugment achieves robustness of 49.87% on the training set, while adversarial training with Cutmix or Mixup still achieves more than 65%.
>
> **Q6:** Why does the combined method (ARD+PRM) sometimes perform worse than any single method (ARD or PRM)?
>
> **A6:** In Table 3, We fixed the hyperparameter ($n_w=10$) for simplicity in all experiments, which may ignore the interaction between these two approaches and make the effect too strong. To reveal the potential of the combined method, we reset the hyperparameter $n_w=5$ (i.e., using fewer epochs for warmup with our combination method), and find it improves the performance as shown in the table below (Taking ViT-S on CIFAR-10 as an example).
>
> |         | vanilla | +ARD  | +PRM  | +both ($n_w=10$) | +both ($n_w=5$) |
> | ------- | ------- | ----- | ----- | ---------------- | --------------- |
> | ROB (%) | 46.37   | 47.12 | 47.22 | 47.19            | 47.32           |
>
> Hope our replies can clear up your doubts and answer your questions. We look forward to the further discussion with you in the Author-Reviewer Discussion period.
>
> **References:**
>
> $[1]$ Namuk Park and Songkuk Kim. How Do Vision Transformers Work?  In ICLR, 2022.
>
> $[2]$ Leslie Rice, Eric Wong, and J. Zico Kolter. Overfitting in adversarially robust deep learning. In ICML, 2020.

---

> ### Author Response · Authors · 2022-08-06
> **Need further clarification?**
>
> Thanks very much for your constructive and detailed comments. We have tried our best to address the concerns and revised our paper accordingly. Is there any unclear point that we should/could further clarify?

---

### Meta-Review · Area_Chair_dHqs · 2022-08-30

**Recommendation:** Accept
**Confidence:** Certain

**Metareview:**

The paper studies how to properly conduct adversarial training on ViTs to obtain adversarially robust models. The paper mainly tried to conduct a comprehensive study on ViT adversarial training and identified several important training heuristics that can improve the robustness of ViTs. Among four reviewers, three consider this as a borderline paper and one strongly supports this work. After several discussions, we think this paper will build a solid foundation for future adversarial robustness studies on ViT models so decide to accept the paper. However, we do hope the authors carefully revise their paper based on the reviewers' suggestions. More specifically, the main concern from reviewers is the correctness issue pointed out by Reviewer 63WY. We hope the authors can carefully check and explain those issues in their final version.

**Award:**

No

---

### Decision · Program_Chairs · 2022-09-14

Accept